# GENERALIZED CONVOLUTIONAL FOREST NETWORKS FOR DOMAIN GENERALIZATION AND VISUAL RECOGNITION

**Jongbin Ryu[1], GiTaek Kwon[1], Ming-Hsuan Yang[2,3,4], Jongwoo Lim[1]** *
[1]Hanyang University, [2]UC Merced [3]Google [4]Yonsei University
`{jongbin.ryu,kwongitack}gmail.com  mhyang@ucmerced.edu`
`jlim@hanyang.ac.kr`

## ABSTRACT

When constructing random forests, it is of prime importance to ensure high accuracy and low correlation of individual tree classifiers for good performance. Nevertheless, it is typically difficult for existing random forest methods to strike a good balance between these conflicting factors. In this work, we propose a generalized convolutional forest networks to learn a feature space to maximize the strength of individual tree classifiers while minimizing the respective correlation. The feature space is iteratively constructed by a probabilistic triplet sampling method based on the distribution obtained from the splits of the random forest. The sampling process is designed to pull the data of the same label together for higher strength and push away the data frequently falling to the same leaf nodes. We perform extensive experiments on five image classification and two domain generalization datasets with ResNet-18, ResNet-50 and DenseNet-161 backbone networks. Experimental results show that the proposed algorithm performs favorably against state-of-the-art methods.

## 1 INTRODUCTION

Random forests have been applied to various problems ranging from object classification (Bosch et al., 2007), object detection (Gall & Lempitsky, 2013), image segmentation (Schroff et al., 2008; Shotton et al., 2008), pedestrian detection (Marin et al., 2013; Tang et al., 2012) to semantic hashing (Qiu et al., 2018). In addition, random forests have been applied to the head pose (Fanelli et al., 2011), landmark (Cootes et al., 2012) and age (Shen et al., 2018) estimation as a regressor and the sparse feature matching problems (Lepetit & Fua, 2006; Ozuysal et al., 2010).

The main reason for the robust performance of random forests is the decision tree ensembles. While each decision tree may achieve mediocre performance, the aggregated random forest performs significantly better and less correlated if the decision trees are heterogeneous. The overall accuracy of the trees can be considered as *strength*, and the heterogeneity of the trees can be measured by *correlation*. In (Breiman, 2001), the upper bound of the generalization error of random forests is expressed in terms of strength and correlation. High strength and low correlation are important properties to minimize the generalization error of a random forest. However, these two conflicting factors make it is difficult to improve strength and lower correlation simultaneously. If the individual decision trees in a random forest are strengthened independently, it is likely for the trees to resemble the strongest tree in the forest, and consequently the correlation of the forest becomes high. To reduce correlation, the decision trees must be in different shapes, and the strength of individual trees would not be as high as the best decision tree.

In this work, we introduce generalized learning for random forests with convolutional neural networks to address these issues. The proposed method iteratively improves the generalized ability of random forests for higher strength and lowers correlation by probabilistic triplet sampling. For a triplet of an anchor, a positive, and a negative sample, the loss function is designed to pull the anchor and the positive closer and to push the anchor and the negative apart. The positive is sampled among the

---

* Corresponding author.

data with the same label with the anchor but in different leaf nodes of the decision trees, and the negative is from the data in the same leaf nodes with the anchor. The former contributes to improving the classification accuracy by positive sampling, whereas the latter discourages the algorithm from constructing similar decision trees by negative sampling. Note that both data points of the same label and of different labels with respect to the anchor can be in the negative training sample set. To directly improve the strength of the random forest only, we may design a method where negative examples are sampled among the data in the same leaf nodes and with different labels from the anchors. However, in practice, it suffers from early saturation and local minima. We describe the details of the proposed learning algorithm and experimental results in the following sections.

The main contribution of the proposed work are summarized as follows:

- We propose a generalization algorithm of random forests with convolutional neural networks. The proposed method minimizes the triplet loss function which 1) encourages to have same-labeled data in different leaf nodes move close and 2) pushes away data points that frequently fall in the same leaf nodes regardless of their class labels.
- We consider both strength and correlation of random forest simultaneously. These two conflicting properties are handled by triple sampling. We demonstrate that the proposed method increases strength while maintaining a correlation in the experiments.
- We show that the proposed algorithm performs well on domain generalization and image recognition against the baseline random forest methods. Furthermore, we show the proposed method performs favorably against state-of-the-art methods for the same tasks.

## 2 PRELIMINARIES

### 2.1 RANDOM FOREST

Since the introduction of random forests (Breiman, 2001), numerous methods have been developed to increase the performance. We discuss recent and relevant methods for vision tasks in this section. In (Gall & Lempitsky, 2013), the Hough transform is used to tally probabilistic voting of detection hypotheses of parts from a random forest for object detection. (Bosch et al., 2007) use local shape and image appearance together to enhance the discriminative strength of the random forest. Visual bag-of-words model and gradient feature are utilized for encoding the shape and local information. In (Zhang & Suganthan, 2014), a method that uses linear discriminant analysis to increase the strength of a decision split is proposed. However, in most cases, increasing the strength does not maximize the performance of the random forest because the correlation also increases. The conflicting factors of strength and correlation in designing decision trees have been studied in recent methods (Rodriguez-Galiano et al., 2012) to analyze the performance of random forest. (Yao et al., 2011) propose to use SVM and image patches to enhance strength while maintaining the correlation of random forest. In this method, the SVM is used as a split function of each node at a decision tree to increase strength and use randomly selected image patches to decrease correlation. While this method performs well on fine-grained classification, it is not clear how this approach can be generalized to other tasks. In contrast, we propose a generalized learning algorithm of random forests that can be applied to various tasks using deep neural networks while considering both strength and correlation.

### 2.2 GENERALIZED ERROR BOUND OF RANDOM FORESTS

(Breiman, 2001) shows that the generalized error $PE^*$ of a random forest is bounded by:

$$PE^* \leq \frac{\bar{\rho}(1-s^2)}{s^2},\tag{1}$$

where $\bar{\rho}$ is the correlation and $s$ is the strength of the random forest. The strength $s$ is the expectation of the margin function $mr(\cdot)$ with respect to a feature $\mathbf{X}$ and its label $Y$:

$$s = E_{\mathbf{X},Y}[\,mr(\mathbf{X},Y)\,], \text{ and}$$

$$f_{mr}(\mathbf{X},Y) = P_\Theta(h(\mathbf{X},\Theta){=}Y) - \max_{j \neq Y} P_\Theta(h(\mathbf{X},\Theta){=}j)\,,$$

where $h(\cdot)$ is a classifier, and $\Theta$ is a random vector used to generate decision trees. In addition, $P_{Theta}(\cdot)$ is the probability described the parameter $\Theta$ for a classifier $h$. The strength represents the

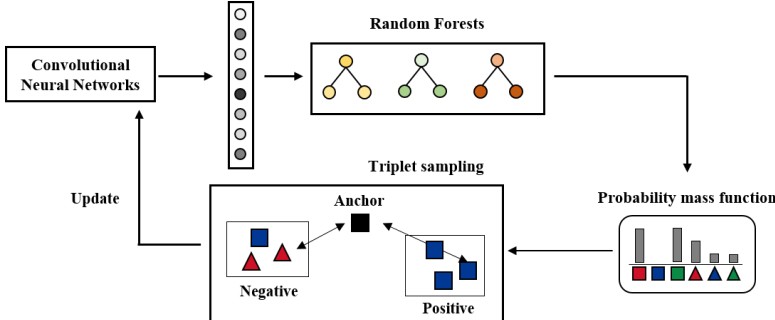

Figure 1: Overall framework of the proposed learning algorithm. Once the random forest is constructed with CNN features, we sample the triplets based on the probability mass function of the split results. The networks are then updated via the loss function of sampled triplets.

expected margin of probability that the random forest makes a correct classification than a wrong classification.

To define the correlation, the raw margin function is used:

$$f_{rmg}(\Theta, \mathbf{X}, Y) = I\left(h(\mathbf{X}, \Theta) = Y\right) - I\left(h(\mathbf{X}, \Theta) = \hat{j}(\mathbf{X}, Y)\right),$$

where $\hat{j}(\mathbf{X}, Y) = \arg\max_{j \neq Y} P_{\Theta}(h(\mathbf{X}, \Theta) = j)$, and $I(\cdot)$ is an indicator function. When $\rho(\Theta, \Theta')$ is the correlation between $rmg(\Theta, \mathbf{X}, Y)$ and $rmg(\Theta', \mathbf{X}, Y)$, the correlation of a random forest $\bar{\rho}$ is the mean value of the correlations over $\Theta$. More details on the generalized error bound of random forests can be found in (Breiman, 2001).

From Eq. 1, it is straightforward to see that high strength of random forest reduces the upper bound of the generalization error if the correlation is suppressed (Breiman, 2001). However, these two conditions cannot be simultaneously satisfied in practice.

## 2.3 TRIPLET LOSS

In general, the loss function of the triplet is defined as:

$$\sum_{(a,p,n) \in S} \max\left(0, \parallel f(a) - f(p) \parallel_2^2 - \parallel f(a) - f(n) \parallel_2^2 + b\right),$$

where $f(a)$, $f(p)$ and $f(n)$ are the feature vectors of the anchor, positive, and negative data for a triplet $(a, p, n)$ in the training set $S$, and $b$ is a user-specified margin. The goal of the training process is to find the best feature $f^*$ that minimizes the loss function. In optimization process, the feature positions of the anchor and the positive sample are pulled closer and those of the anchor and the negative sample are pushed away.

The triplet loss function has been used in numerous applications including face recognition (Parkhi et al., 2015; Schroff et al., 2015), image retrieval (Zhao et al., 2015), person re-identification (Cheng et al., 2016; Zhang et al., 2016), and metric learning (Norouzi et al., 2012; Wang et al., 2014), to name a few. The triplet loss function uses both positive and negative samples at the same time, thus achieves improved performance. In the existing methods, positive samples are the data of the same labels with the anchor, and the negative samples are the data with different labels. For face recognition, person identities are the labels, and for person re-identification, the tracklet determines the positive and negative sample sets. Minimizing the triplet loss then gathers the same-labeled data together and separates differently-labeled data apart on the learned feature space, so that a classifier can easily partition the data.

Compared to the above approaches, our focus on how to sample the effective triplets to improve both classification ability and heterogeneity of random forests. We show that in random forest training simply clustering data according to the labels does not bear the best result. Since it increases both the strength and correlation of the random forest, and we present the supportive experimental results.

---

**Algorithm 1** Learning algorithm for generalized convolutional forest networks.

---

**Input:** Training Image $I$, Class label $Y$
**Output:** Generalized Neural Networks $N^*$

1: **for** $i \leftarrow 1$ to maximum iterations **do**
2:     $F' \leftarrow N_i(I)$
3:     Construct decision trees from $F'$ and $Y$
4:     Construct $P^p$ and $P^n$ from split results by the decision trees
5:     $S \leftarrow$ Sample triplets by $P^p$ and $P^n$
6:     $N_{i+1} \leftarrow$ Update by minimizing triplet loss on $S$

---

## 2.4 DOMAIN GENERALIZATION

Generalizing models learned from one domain to another is an important topic in machine learning and computer vision. Learning deeply connected neural networks with a large-scale dataset helps improve the generalized ability such that CNN features trained on the ImageNet dataset are used as the *generic* representation for various visual domains (Sharif Razavian et al., 2014). However, it is still difficult to adapt a model to different domains when large domain gaps exist. In addition, it is even more challenging if we do not have any data from the target domain, or if we have a mixture of source and target domains data. Hence, numerous domain generalization methods have recently been developed to tackle this problem. Several approaches have been developed including regularization with meta-learning (Balaji et al., 2018), domain-invariant conditional learning (Li et al., 2018b), adversarial back-propagation (Li et al., 2018a), and episodic training algorithm (Li et al., 2019).

## 3 PROPOSED ALGORITHM

In this section, we formulate the problem of improving strength and maintaining the correlation of random forests and describe the generalized feature learning algorithm to address these two factors simultaneously.

We first present a feature learning algorithm which only considers strengthened features of random forests. The modified feature learning algorithm on CNNs is then introduced to achieve high strength and low correlation at the same time for the proposed Generalized Convolutional Forest Network (GCFN). The overall framework of the proposed GCFN method is shown in Fig. 1 and Algorithm 1.

## 3.1 STRENGTHENED LEARNING ALGORITHM OF RANDOM FORESTS

The classification result of an input $\mathbf{x}$ to a random forest is determined as:

$$c^* = \arg\max_c \frac{1}{T} \sum_t P(c|\lambda_t(\mathbf{x})),$$

where $c$ is the label, $T$ is the number of decision trees, and $\lambda_t(\mathbf{x})$ denotes the leaf node of a tree $t$ into which $\mathbf{x}$ falls. Here, $P(c|\cdot)$ is the conditional probability of $x$ belonging to class $c$. In other words, $\lambda$ can be thought of as a mapping function from $\mathbf{x}$ to a probability distribution on the label space. To maximize the strength of a decision tree, each leaf node should only contain data with a single label, i.e., the distribution should have a single entry with probability one and others with zeros. Intuitively if the data with same labels are converged together in the space, the leaf nodes are more likely to contain single labeled data, and we can design a triplet loss to maximize the data clustering in the learned space.

The networks are updated via the probabilistic triplet sampling. To construct a triplet sample set, we randomly sample anchors $\{\mathbf{a}_i\}$, and for each anchor $\mathbf{a}_i$ one positive sample $\mathbf{p}_i$ and one negative sample $\mathbf{n}_i$ are randomly drawn according to the probability mass functions (PMFs) for positive and negative pools of the anchor, i.e., $\mathbf{p}_i \sim P^p(\mathbf{a}_i)$ and $\mathbf{n}_i \sim P^n(\mathbf{a}_i)$. The positive pool consists of the data with the same label with an anchor $\mathbf{a}$ but in different nodes of the tree, and the negative pool contains the data in the same node but with different labels. Given the anchor $\mathbf{a}$, the PMFs of the positive and negative samples are defined by:

$$P^p(\mathbf{x}; \mathbf{a}) \propto \sum_t I\big(\lambda_t(\mathbf{x}) \neq \lambda_t(\mathbf{a}) \wedge y(\mathbf{x}) = y(\mathbf{a})\big), \text{ and}$$

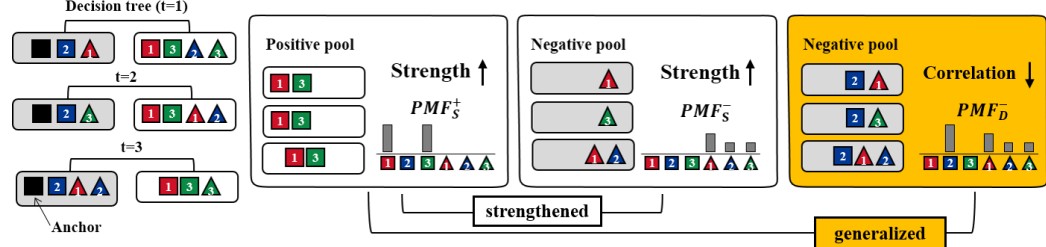

Figure 2: Probability mass function for sampling triplets. Probability mass functions for positive and negative samples ($P^p$ and $P^n$ respectively) are constructed by sample distribution with the anchors in leaf nodes. Squares and triangles represent training data in leaf nodes of the decision trees, and the shapes represent their labels. For an anchor (black square), the positive pool contains the data in different leaf nodes and with the same label (red and blue squares), and the negative pool contains either the data in the same leaf node with the same or different labels. Probability mass functions are the normalized histogram of the positive and negative pools.

$$P^n(\mathbf{x}; \mathbf{a}) \propto \sum_t I\big(\lambda_t(\mathbf{x}) = \lambda_t(\mathbf{a}) \wedge y(\mathbf{x}) \neq y(\mathbf{a})\big),$$

where $I(\cdot)$ is an indicator function returning 1 if true and 0 otherwise, and $y(\mathbf{x})$ returns the label of $\mathbf{x}$. Both PMFs need to be normalized to sum to one. The networks $N$ are updated using the triplet samples $S$ by minimizing the loss function Eq. 2.

$$L = \sum_{(\mathbf{a}, \mathbf{p}, \mathbf{n}) \in S} \| N(\mathbf{a}) - N(\mathbf{p}) \|_2^2 - \| N(\mathbf{a}) - N(\mathbf{n}) \|_2^2. \tag{2}$$

The proposed random forest on the strengthened feature space shows improved performance compared to that of the canonical feature space, but the improvement saturates quickly and sometimes it fails to converge. The reasons can be attributed to:

- the correlation of the random forest increases rapidly along with strength improvement, and thus the gain in overall performance is limited, as well as
- the optimization process often falls into local minima.

The data points with the same labels are pulled together and naturally, individual decision trees become stronger, but at the same time, the decision trees become similar. The growth of correlation is apparent because if two same-labeled data is in one leaf node, they are likely to stay close and belong to the same leaf nodes in the next iteration. In practice, the local minimum issue affects performance more critically. The update process of learning strengthened feature is analogous to the steepest descent algorithm in optimization, in the sense that both positive and negative samples concentrate on the strengthening random forests.

## 3.2 GENERALIZED LEARNING ALGORITHM OF RANDOM FORESTS

To alleviate the above-discussed issues, we present a triplet sampling method. As the positive sampling rule is effective in enhancing the strength, we design the negative sampling rule to deal with the correlation and the local minima, as shown in Fig. 2. The PMF for negative sampling is defined as:

$$P^n(\mathbf{x}; \mathbf{a}) \propto \sum_t I\big(\lambda_t(\mathbf{x}) = \lambda_t(\mathbf{a})\big).$$

The role of negative sampling is two-fold. First, it prevents the correlation of the random forest growing quickly. If two data points belong to the same nodes of many decision trees (which causes high correlation), they are likely to be sampled as the negative examples and pushed away from each other. The probability of these data points belonging to the same nodes in the next iteration becomes smaller, and the correlation of decision trees decreases.

Second, it helps prevent the update process stuck in local minima and contributes to achieving higher strength and classification accuracy than strengthened feature learning algorithm. Since the negative

sampling diffuses the data, it operates in a way similar to the regularization term in optimization for dealing with local minima issues. Hard negative examples can cause the learning process to fall into a local minimum, and a recent method (Schroff et al., 2015) suggests to exclude the negative data points too close to anchors from sampling. In this work, strengthened feature learning gives the high probability for negative sampling to the hard negative examples since they will be in the same nodes in most decision trees, but it is difficult to detach them from the anchors in the strengthened feature space. In the proposed triplet sampling strategy, the weight of the hard negative examples is spread out to other positive samples in the same nodes. Thus it learns generic feature space for random forests without getting stuck in local minima. The proposed GCFN is designed to address the above-discussed issues. In the following, we present the various experimental validations for the proposed method.

## 4 EXPERIMENTAL RESULTS

In this section, we present experimental results of the proposed method for domain generalization and visual recognition tasks using the same backbone networks. In comparison with random forest based on different spaces, the GCFN method performs favorably in all classification tasks. We also present detailed discussions on the strength and correlation of the trained random forests and the properties of the optimized learned space. Finally, we show the performance comparison of proposed random forests with the state-of-the-art method in such tasks. Due to space constraints, the details of the experimental settings are discussed in the appendix.

Table 1: Comparison of random forests on canonical (can), strengthened (str) and generalized (gen) space. We measure the classification accuracy with $T = 1, 10, 50$ for three datasets. In all datasets, random forests on the generalized feature space perform well. The best result in each number of trees is marked as bold. It is worth noticing that when the number of trees is 1, the random forest with the strengthened space performs better. As the tree grows, the correlation of this random forest increases, and the proposed method with the generalized space performs better.

| Space | MIT Indoor | | | Scene-15 | | | 4D-Light | | |
|-------|------|------|------|------|------|------|------|------|------|
| | 1 | 10 | 50 | 1 | 10 | 50 | 1 | 10 | 50 |
| can | 26.1 | 54.6 | 65.6 | 61.7 | 83.4 | 88.1 | 40.0 | 65.6 | 73.6 |
| str | **48.4** | 66.9 | 70.8 | **84.3** | 88.6 | 89.3 | **68.3** | 73.9 | 76.9 |
| gen | 46.0 | **69.3** | **74.0** | 80.1 | **90.2** | **91.6** | 66.9 | **78.3** | **79.7** |

Table 2: Comparison of random forests on canonical (can), strengthened (str) and generalized (gen) space. We measure the classification accuracy with $T = 50$ for five datasets. Random forests with the generalized space achieve the best result in all settings.

| Network | Space | MIT-Indoor | | Scene-15 | | 4D-Light | | DTD | | Stanford-Dog | |
|---------|-------|------|------|------|------|------|------|------|------|------|------|
| | | F | S | F | S | F | S | F | S | F | S |
| ResNet | can | 65.6 | 72.5 | 88.1 | 91.7 | 73.6 | 79.7 | 65.1 | 70.9 | 85.1 | 86.2 |
| | str | 70.8 | 71.7 | 89.3 | 90.2 | 76.9 | 78.3 | 70.3 | 71.4 | 83.4 | 84.1 |
| | gen | **74.0** | **75.7** | **91.6** | **92.4** | **79.7** | **81.1** | **71.5** | **72.5** | **85.7** | **86.7** |
| DenseNet | can | 63.7 | 67.8 | 89.2 | 91.2 | 75.6 | 77.2 | 67.6 | 66.2 | 77.7 | 83.1 |
| | str | 71.9 | 72.6 | 89.8 | 90.4 | 77.8 | **81.4** | 69.8 | 70.9 | 83.0 | 84.8 |
| | gen | **74.6** | **77.2** | **91.7** | **92.0** | **79.2** | 81.1 | **70.7** | **72.2** | **85.4** | **86.8** |

## 4.1 EVALUATION WITH BASELINE FORESTS

We first compare the proposed feature learning algorithms for random forests in Fig. 3 of the appendix. In most cases, random forest with the generalized features outperforms that of the strengthened features in strength, correlation, $PE^*$, and classification accuracy. Although the proposed feature learning method using the split results breaks the independence of the decision trees, such metrics are still valid to demonstrate the implications of the proposed method. Since the split results are

used only for the feature learning but are not used when building each decision tree from the learned feature.

Although random forests on the strengthened feature space outperform canonical random forests in classification accuracy, the performance reaches to a plateau quickly. On the other hand, the performance of random forests with the generalized feature space increases steadily in classification accuracy and strength, and the correlation is maintained at the lower levels than that of strengthened feature space. Since the strengthened feature learning method aims to improve the strength only, the strength grows rapidly at the beginning, but the correlation also gets higher. After several iterations, this approach falls into local minima, but the generalized feature learning method continuously reduces the upper bound of generalization error and reaches much higher accuracy. It is worth noticing that the strength of the generalized feature space is often much higher than that of strengthened feature space. This can be explained by that feature the learning process with the strengthened feature space is easily caught in local minima, whereas the proposed method can escape or avoid the local minima. More importantly, the correlation of the random forests on generalized features stays similar to that of canonical feature space while its strength is much higher. Hence the generalization error $PE^*$ of the proposed method is much smaller. It can also be elucidated by the overfitting problem of bias and variance dilemma. The proposed positive sampling scheme reduces bias, while negative sampling process alleviates the overfitting problem. Although the random forest reduces the variance from the bagging process, it can quickly be overfitted if the positive sampling scheme reduces the bias too fast. The negative sampling process, on the other hand, regularize the steepest reduction of the bias; and thus, the proposed method enjoys the merits of the random forest with low bias and variance for the classification task.

Table 1 and 2 summarize the results of canonical and proposed random forests on various deep features, number of trees, split functions, and different classification datasets. In all cases, the proposed random forest method on the generalized feature space performs significantly better than the canonical random forest schemes. In addition, the experimental results show that the proposed method is not designed for a specific task, but can be applied to numerous classification tasks. This shows the generalization ability of the proposed method, along with the experimental results of domain generalization in the next subsection.

## 4.2 EVALUATION WITH STATE-OF-THE-ART METHODS

We evaluate the performance of the GCFN method with previous state-of-the-art methods for domain generalization and various visual recognition tasks. Here we train the GCFN method with the generalized feature learning algorithm and the split function 'S'. We use the depth value depending on the number of training samples and sufficient iterations of the learning stage to maximize the performance of the GCFN.

The domain generalization task is evaluated in Table 3 and 4. We compare GCFN with recent domain generalization methods such as D-SAMs (D'Innocente & Caputo, 2018), DANN (Ganin et al., 2016), MetaReg (Balaji et al., 2018), MMD-AAE (Li et al., 2018a) and Epi-FCR (Li et al., 2019). The proposed GCFN algorithm performs favorably against state-of-the-art methods for both datasets. The results show that the proposed GCFN algorithm learns the generic distribution to classify unseen domain data.

The results of Table 5-9 also show the GCFN works well compared to state-of-the-art methods in visual recognition tasks. We compare the classification accuracy with recent state-of-the-art results from FV-CNN (Cimpoi et al., 2015), DeepTen (Zhang et al., 2017), DEP (Xue et al., 2018) and DFT (Ryu et al., 2018) for MIT-Indoor, 4D-Light and DTD datasets in Table 5, 6 and 8. For these experiments, the multiscale training scheme is applied to the ResNet-50 backbone networks. The details on how to utilize the multiscale scheme and backbone networks are slightly different for each other, but we expect each to use the best settings for their method. The GCFN method is trained on the ResNet-152 backbone networks for the Scene-15 data set in comparison with the ResNet+weighted_layout (Weng et al., 2016) scheme in Table 7. Although they do not use the multiscale scheme, the spatial pyramid pooling is applied to use the spatially multi-level features. The PC(ResNet-50), PC(Densenet-161) (Dubey et al., 2018) and MAMC(ResNet-50) (Sun et al., 2018) methods are evaluated with the GCFN(ResNet-50) algorithm in Table 9. Overall, the proposed GCFN method performs favorably against state-of-the-art methods. Since the random forest has been one of the most widely used classifiers to the visual data, we show the generic performance of the proposed GCFN in the five datasets of three visual domains. These experimental results demonstrate the generalization ability of the proposed algorithm.

Table 3: Experiments on the OfficeHome dataset with the ResNet-18 backbone.

| Method | Art | Clipart | Product | Real-world | Average |
|---|---|---|---|---|---|
| Deep All (feat.) | 52.7 | **48.4** | 71.4 | 71.5 | 61.0 |
| Deep All | 55.6 | 42.4 | 70.3 | 70.9 | 59.8 |
| D-SAMs | 58.0 | 44.4 | 69.2 | 71.5 | 60.8 |
| GCFN | **61.9** | 44.8 | **75.2** | **76.8** | **64.7** |

Table 4: Experiments on the VLCS dataset with the AlexNet backbone.

| Method | Pascal | Labelme | Caltech | Sun | Average |
|---|---|---|---|---|---|
| DANN | 66.4 | 64.0 | 92.6 | 63.6 | 71.7 |
| MetaReg | 65.0 | 60.2 | 92.3 | 64.2 | 70.4 |
| MMD-AAE | 67.7 | 62.6 | **94.4** | 64.4 | 72.3 |
| Epi-FCR | 67.1 | **64.3** | 94.1 | 65.9 | 72.9 |
| GCFN | **73.8** | 61.7 | 93.9 | **67.5** | **74.2** |

Table 5: Experiments on the MIT-Indoor dataset.

| Method | DeepTen | DFT | DFT+ | GCFN |
|---|---|---|---|---|
| Acc | 76.2 | 78.6 | 80.2 | **80.3** |

Table 6: Experiments on the 4D-Light dataset.

| Method | FV-CNN | Deep-Ten | GCFN |
|---|---|---|---|
| Acc | 77.6 | 81.4 | **82.2** |

Table 7: Experiments on the Scene-15 dataset.

| Method | ResNet+SVM | ResNet+wl | GCFN |
|---|---|---|---|
| Acc | 92.3 | **94.5** | 94.3 |

Table 8: Experiments on the DTD dataset.

| Method | FV-CNN | Deep-Ten | DEP | GCFN |
|---|---|---|---|---|
| Acc | 72.3 | 69.6 | 73.2 | **76.8** |

Table 9: Experiments on the Stanford-Dog dataset. The results are acquired on the ResNet-50 and DenseNet-161 for PC, MAMC, and GCFN.

| PC (ResNet) | PC (DenseNet) | MAMC (ResNet) | GCFN (ResNet) | GCFN (DenseNet) |
|---|---|---|---|---|
| 73.4 | 83.6 | 84.8 | 86.7 | **86.8** |

## 5 CONCLUSIONS

In this paper, we propose the GCFN method which learns the generalized feature space iteratively such that the discrimination strength of each tree classifier is increased while the correlation is suppressed. The proposed learning algorithm uses the triplet sampling on the probability distributions of split results of the decision trees. The data with the same label with the anchor but in different nodes are likely to be positive samples to increase strength, and the data in the same nodes with the anchor be negative samples to suppress correlation and diffuse data to avoid falling in local minima. We experimentally show that the proposed method outperforms baseline random forests on various experiments. Furthermore, the proposed algorithm performs favorably against state-of-the-art methods for domain generalization and visual recognition tasks.

## ACKNOWLEDGEMENTS

This work was supported by the National Research Foundation of Korea(NRF) grant funded by the Korea government(MSIT) (NRF-2019R1A4A1029800), Basic Science Research Program through the National Research Foundation of Korea(NRF) funded by the Ministry of Education (NRF-2017R1A6A3A11031193), and NSF CAREER Grant No.1149783.

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

# A  APPENDIX

We detail the experimental settings and performance evaluation of random forests on different feature spaces in the appendix.

We validate the effectiveness of the proposed method for domain generalization and visual recognition tasks. We conduct experiments on various visual recognition tasks using three domains, such as scene, texture, and fine-grained images. Domain generalization experiments are carried out to demonstrate the effectiveness of the generalization ability of the proposed method.

We use five datasets for the three visual recognition domains. The MIT-Indoor (Quattoni & Torralba, 2009), Scene-15 (Lazebnik et al., 2006; Oliva & Torralba, 2001), 4D-Light (Wang et al., 2016), DTD (Cimpoi et al., 2014), and Stanford-Dog (Khosla et al., 2011) datasets are utilized for the scene, texture, and fine-grained domains. We use the ResNet (He et al., 2016), DenseNet (Huang et al., 2017) and AlexNet (Krizhevsky et al., 2012) trained on ImageNet dataset (Russakovsky et al., 2015) as backbone networks for visual recognition tasks. We use the standard protocols of training and test data split. In addition, we include two datasets for evaluation of the domain generalization. The Office-Home (Venkateswara et al., 2017) with ResNet-18 and VLCS (Fang et al., 2013; Torralba & Efros, 2011) with AlexNet methods are used for evaluation against the state-of-the-art schemes. We use three source domains as training data and the other one target domain as test data for Office-Home. For VLCS, the split of 70% source with 30% target data for all domains is used in the experiments.

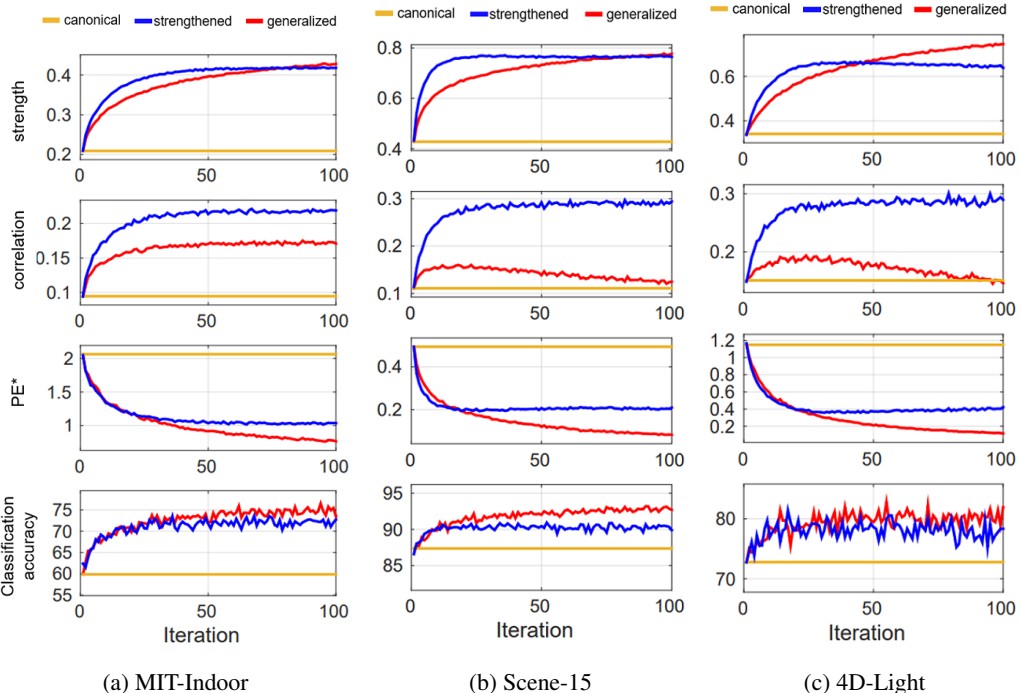

(a) MIT-Indoor      (b) Scene-15      (c) 4D-Light

Figure 3: Performance evaluation of random forests constructed on canonical, strengthened and generalized feature space. The iteration on each figure is equal to the epoch of the network training. Random forest on strengthened feature space saturates quickly but on generalized space improves stably. The strength of the random forests learned from the strengthened space is similar to or higher than that of the one with the generalized space, but the correlation is much lower. As a result, the random forest learned from the generalized space provides a much lower upper bound of generalized error and higher accuracy of in all test cases.

Table 10: Comparison of random forests on the SVHN (Netzer et al., 2011) and CIFAR-100 (Krizhevsky & Hinton, 2009) dataset. We used the LeNet (LeCun et al., 1998; LeCun, 2015) based backbone network.

| | SVHN | | | CIFAR-100 | |
|------|------|------|------|------|------|
| can | str | gen | can | str | gen |
| 93.0 | 94.7 | 94.8 | 52.0 | 52.5 | 54.8 |

We additionally evaluate the proposed method on the SVHN and CIFAR-100 datasets in Table 10. We train the random forest on the canonical feature space using pre-trained CNN models. It also shows random forests on the generalized features space perform better than two baseline random forests.

Each node of decision trees in a random forest has a split function to determine which child to follow. The split function is usually defined as a threshold function on a feature dimension that maximizes the entropy (Shotton et al., 2008), and in our experiments, we indicate such split function by the 'F'. Recently split functions such as PCA (Wold et al., 1987), LDA (Cohen et al., 2014) or SVM (Schölkopf et al., 2002) are introduced for better performance (Zhang & Suganthan, 2014; Yao et al., 2011). In this work, we use the SVM split function in our implementation, and they are marked with the 'S'. The balanced node split learning of (Ryu et al., 2020) is utilized for both 'F' and 'S' split functions. We use random $n$-dimensional subspaces of the input vector for node-wise SVM training and $n$ is set depending on the size of the training data. Each input image is resized to $224 \times 224$ for the single-scale and $224 \times 224$, $288 \times 288$, $352 \times 352$, $224 \times 224$ and $512 \times 512$ for the multi-scale setting for visual recognition tasks except the Stanford-Dog dataset. Similar to previous fine-grained recognition studies, we use $448 \times 448$ input images for the Stanford-Dog dataset in the experiments.

We use 300 epochs with $5 * 10^{-3}$ learning rate, weigh decay with $9 * 10^{-4}$ for training. Random forests are created at every 3 epochs to reduce the overhead. Once we build a random forest, the neural networks are updated for 3 epochs based on the split result of the random forest. The training time overhead due to the random forests is 15 seconds for MIT Indoor dataset with ResNet-50, which results in only 25 minute overhead in the whole learning process. For the fast training, we use 50 trees with the split function of 'F' for the network update.

