# OpenReview forum: "Generalized Convolutional Forest Networks for Domain Generalization and Visual Recognition"
_ICLR.cc/2020/Conference — Accept (Poster)_

### Official Review · AnonReviewer1 · 2019-10-21
**Official Blind Review #1**

**Rating:** 6

**Review:**


The paper proposes a scheme to learn representations and a random
forest model in a closely coupled manner, where the representations
are learned to simultaneously improve the predictive performance of
the individual trees in the random forest and reduce the correlation
between these trees. This is achieved via a novel triplet sampling
scheme where the positive samples are used to improve the predictive
performance of the individual trees, while the negative samples are
used to reduce the correlation between the trees. The empirical
results indicate the improvement of the proposed scheme over random
forests learned over the original representations across various
datasets.

The manuscript is well presented and I am leaning towards accept, one
major issue with this manuscript is the number of pages which crosses
the 8 page recommended limit in the ICLR CfP. Given that higher
standard, I feel that there are multiple minor issues which should be
addressed.

- It is not clear why the proposed scheme is especially well suited
  for "domain generalization" and "visual recognition" and not for
  general supervised classification tasks (like those on CIFAR, SVHN,
  etc). This needs to be clarified.
- The theoretical results for random forest rely on the independence
  of the individual trees. Given that the trees in the random forest
  are no longer independent in the proposed scheme, the upper bound in
  Equation (1) may no longer be valid. While this does not affect the
  empirical performance, it might be good to discuss the theoretical
  implications of the proposed scheme.
- It is not clear why the curves for the canonical features in Figure
  3 not improving with the number of iterations (which corresponds to
  number of trees to the best of my understanding). The results in
  Table 1 do indicate improvement in the performance with canonical
  features, putting Figure 3 and Table 1 at odds.


Minor:

- The individual trees in a random forest are supposed to have low
  bias but high variance, and the heterogeneity between the trees is
  supposed to squash the variance with the bagging. The introduction
  mentions that the individual trees in the forest are "less
  biased". Maybe this is because of our different definitions of bias
  but any clarification here would be helpful.


**Experience Assessment:**

I have read many papers in this area.

**Review Assessment: Checking Correctness Of Derivations And Theory:**

N/A

**Review Assessment: Checking Correctness Of Experiments:**

I carefully checked the experiments.

**Review Assessment: Thoroughness In Paper Reading:**

I read the paper at least twice and used my best judgement in assessing the paper.

---

> ### Author Response · Authors · 2019-11-14
> **Response to Reviewer 1**
>
> We thank Reviewer 1 for the helpful comments. We answer to the comments of Reviewer 1.
>
> 1. General supervised classification tasks
> To demonstrate the generalization ability, we evaluate the proposed GCFN in domain generalization and visual recognition tasks. This is because domain generalization itself evaluates the generalization performance on different domain settings of training and test sets. Visual recognition of various domains evaluates recognition performance in terms of generalization due to multiple test domains.
>
> We will add the experiments with the CIFAR and SVHN data sets to demonstrate the effectiveness of the GCFN. Due to the short rebuttal period, the experiment results will be added in the final version of this paper.
>
> The proposed method is not only suitable for domain generalization and visual recognition of the paper, but can be extended to the general recognition tasks on the CIFAR and SVHN datasets.
>
> 2. It is correct that the proposed method iteratively learns a generalized feature distribution based on the split results of a random forest. Thus, the learned feature distribution is not independent of each decision tree. However, when creating a new decision tree in a random forest, it does not take into account the splitting results of previously trained decision trees. In other words, the feature learning algorithm includes a process that depends on the partition results of the decision tree, but there is no dependency process for the train decision tree itself. Thus, we use a metric of the upper bound of generalization error for theoretical implications.
>
> As suggested by the reviewer, it is worth studying more because learning methods break the independence of input functions. However, we show that the upper bound of generalization error and actual experimental results are almost the same, and the results are meaningful for showing the generalization ability of the proposed method.
>
> 3. We apologize for the confusion of the representations in Figure 3 and Table 1. In fact, the x-axis in Figure 3, labeled 'iteration', represents the ‘epoch’ of the network learning procedure, but not the number of trees. Thus, there is no network learning about canonical features, which does not improve the accuracy with regard to the ‘epoch’ for the canonical features. We clarify that the ‘iteration’ in Figure 3 denotes the ‘epoch’ in the revised paper.
>
> 4. We use the word “bias” to indicate the relationship between trees. The term “less bias” means “less correlated to each other trees”. We have revised the sentence as below.
>
> While each decision tree may achieve mediocre performance, the aggregated random forest performs significantly better and less correlated if the decision trees are heterogeneous.
>
> The correction can be found on page 1 of the revised paper.

---

### Official Review · AnonReviewer3 · 2019-10-23
**Official Blind Review #3**

**Rating:** 3

**Review:**

This paper proposes a method for generalized image recognition based on random forest, that use directly the features extracted by the backbone, that assign pseudo-labels to the data. The learning is performed with a triplet loss adapted for better generalization.
Decision: weak reject
Motivation: the method is incremental and presented in general in a clear way and easy to follow, the authors present a simple but interesting trick to make the triplet loss more effective on a random forest in the case of generalization to a new unlabeled dataset. This method looks incremental to me because it is addressing the problem of pseudo-labelling for learning on a new dataset and instead of using confidence measures uses a random forest to assign labels.
The experimental section of the paper is a bit confusing because is not clear if the results presented are with comparable network (e.g. ResNet18) like the cited state-of-the-art papers, from further readings I am confident the autors compared fairly with similar architectures. Authors should perhaps stress they compare with state-of-the-art in fair condition to avoid confusion as in my case. How much is the overhead of building the random forest for each iteration of the learning (algorithm 1), a more detailed analysis on this is useful for understanding the method. Could this method be used to train a network from scratch on an unlabeled data or on data with noisy labels? How did the authors choose the T decision trees, is there any ablation study, general practice or euristics behind the choice of 1,10,50? The comparison with state-of-the-art Tab 3 and Tab 4 shows that for some datasets other techniques are better, did the authors draw some conclusions from that? Comparing Tab 3 and 4 with Tab 5/6/7/8/9 looks like this method can work but only in the case of much bigger network like ResNet50 and DenseNet 161 which can limit its use for high resources (computing power) cases.
Replicability: I think with improvements in the experimental section the method results can be replicated. At the moment it lack many details like learning rates, epoch of training and other useful information that are useful.
Minor: there are two lines out of the 9 page limit

**Experience Assessment:**

I have published in this field for several years.

**Review Assessment: Checking Correctness Of Derivations And Theory:**

I assessed the sensibility of the derivations and theory.

**Review Assessment: Checking Correctness Of Experiments:**

I carefully checked the experiments.

**Review Assessment: Thoroughness In Paper Reading:**

I read the paper thoroughly.

---

> ### Author Response · Authors · 2019-11-14
> **Response to Reviewer 3**
>
> We thank Reviewer 3 for the helpful comments. We answer to the comments of Reviewer 3.
>
> 1. Backbone networks
> We compare the proposed algorithm with the state-of-the-art methods using the same backbone network such as ResNet-18 (Table 3), AlexNet (Table 4), ResNet-50 (Table 5, 6, 8 and 9), ResNet-152 (Table 7) and DenseNet-161 (Table 9).
>
> In Table 9, we use two backbone networks (ResNet-50 and DenseNet-161) for comparisons. Since pairwise confusion (PC) [1] report results from both backbone networks, we compare the proposed method with PC on both for fair comparisons.
>
> We also revised the paper to clarify the backbone networks on page 6 and abstract.
>
> [1] Abhimanyu Dubey, Otkrist Gupta, Pei Guo, Ramesh Raskar, Ryan Farrell, and Nikhil Naik. Pairwise confusion for fine-grained visual classification. In European Conference on Computer Vision, pp. 70–86, 2018.
>
> 2. Overhead of each iteration
> We measured the overhead of training time for random forests.
>
> As stated in answer 3, we use 300 epochs and random forests are trained at every 3 epochs for reducing the overhead.
>
> The time for training a random forest is about 15 seconds (MIT Indoor dataset with ResNet-50), which results in only 15sec X 300 epochs / 3 = 25 minute overhead in the whole learning process.
>
> 3. The number of trees
> We compare the performance of random forests constructed on canonical, strengthened, and generalized feature space with T=1,10 and 50 in Table 1. As shown in [2] there is little change in performance above 64 trees for a random forest,  we measure the performance change from 1 to 50 in a way similar to [2]. Table 1 shows the ablation study of the performance with regard to the number of trees.
>
> [2] Oshiro Thais Mayumi, Perez Pedro Santoro, Baranauskas Jos´e Augusto, How many trees in a random forest?. In International workshop on machine learning and data mining in pattern recognition, pp. 154–168, 2012
>
> 4. Some results in Table 3 and 4
> There are some cases that the proposed GCFN does not outperform the state-of-the-art methods. However, the GCFN outperforms them in terms of average accuracy for each dataset, which shows the effectiveness of GCFN. It is more important to analyze the overall average performance of all cases to validate the generalization ability rather than one or two cases.
>
> 5. Validation on small networks
> We perform some additional experiments on the AlexNet [1] which is relatively small and fast networks compared to ResNet-50 and DenseNet-161.
>
> We compare the random forests on canonical features as the baseline with the proposed GCFN on the DTD, MIT Indoor and Scene-15 datasets.
>
> The results also confirm that GCFN also performs well with relatively small networks.
>                             DTD                      MIT Indoor                  Scene-15
>                   T=1   T=10  T= 50    T=1  T=10  T= 50       T=1   T=10  T= 50
> Base          44.9   60.4  62.8       39.9  59.2   62.0         79.9  88.5   88.9
> GCFN        55.0   63.1  64.2       51.6  62.1   63.1         84.7  88.7   89.5
>
> [1] Alex Krizhevsky, Ilya Sutskever, and Geoffrey E Hinton. Imagenet classification with deep convolutional neural networks. In Neural Information Processing Systems, 2012
>
> 6. Implementation details
> We add the implementation details about the learning rate, batch size, number of epochs, and training process in Page 12 of the paper as below.
>
> We use 300 epochs with 5*10^(-3) learning rate, weigh decay with 9*10^(-4) for the network training. Random forests are created at every 3 epochs to reduce the overhead. Once we build a random forest, the neural networks are updated for 3 epochs based on the split result of the random forest. The training time overhead due to the random forests is 15 seconds for MIT Indoor dataset with ResNet-50, which results in only 25 minute overhead in the whole learning process. For the fast training, we use 50 trees with the split function of `F' for the network update.
>
> 7. Page limit
> We fix the page limit issue by moving some parts to the appendix.

---

### Official Review · AnonReviewer2 · 2019-10-26
**Official Blind Review #2**

**Rating:** 6

**Review:**

The paper aims to improve the random forest performance by iterative constructing & feeding more powerful features that are to be used by the random forest learning processing where a random subset of features are chosen from the current feature pool when making growing/stopping decision at the current split node. The idea is new and interesting, and its usefulness has been empirically shown. On the other hand, it is not clear how this additional procedure would do with the good properties of RFs such as less subjective to overfitting and bias. It would be very helpful if the paper could shred some lights in this regard.


**Experience Assessment:**

I have published one or two papers in this area.

**Review Assessment: Checking Correctness Of Derivations And Theory:**

I assessed the sensibility of the derivations and theory.

**Review Assessment: Checking Correctness Of Experiments:**

I assessed the sensibility of the experiments.

**Review Assessment: Thoroughness In Paper Reading:**

I read the paper at least twice and used my best judgement in assessing the paper.

---

> ### Author Response · Authors · 2019-11-14
> **Response to Reviewer 2**
>
> We thank Reviewer 2 for the helpful comments. We answer to the comments of Reviewer 2.
>
> We revised the paper to explain the obvious benefits of the proposed method in terms of overfitting, bias and generalization in Page 7.
>
> A detailed description of this topic follows.
>
> It is known that large bias and variance cause overfitting problems. The model generalization reduces the overfitting problem by regularizing the learning model. A well-trained random forest is known to have low bias and variance to generalize the classification task. To have low bias of a random forest, a good practice is to learn the decision tree very deeply. For low bias in each tree, the bagging strategy of a random forest reduces the variance for the final prediction. Therefore, in general, random forests can have low bias and variance.
>
> However, there is still bias and variance dilemma in training decision trees. Making each decision trees very deep increases the variance of the decision tree. Suppose that if tree depth is one, there are only two decisions that the tree can make, which leads to lower variance. However, if the tree depth is ten, the number of decision cases will be about one thousand, resulting in a much higher variance. Therefore, if each decision tree has too high variance due to the depth, a random forest can still have a large variance even after the bagging process. To mitigate this problem, we introduce the GCFN, which constructs very low bias where the variance is not too high. We use the strength and correlation terms to theoretical analyze the overfitting problem due to the bias and variance dilemma. The positive sampling strategy of our method contributes to increasing the strength, and thus each tree has a low bias value. On the other hand, the negative sampling scheme regularizes the rapid increase of the strength, which can cause the overfitting problem while reducing the correlation. We measure the degree of model overfitting in terms of the upper bound of generalization error in the paper. We show that the proposed sampling strategy reduces the upper bound of the generalization error, and therefore, the GCFN can provide the generalized performance of various visual recognition tasks and avoid the overfitting problem even with very low bias.
>
> We also modify the word “bias” in the introduction section to better describe our intention.
> We mean to use the word “bias” to describe that each tree looks similar.  The word “bias” does not have the same meaning as the above discussion on the bias and variance dilemma. The revised text can be found on page 1 of the paper.

---

### Decision · Program_Chairs · 2019-12-19

**Decision:**

Accept (Poster)

**Comment:**

The authors introduce an approach to learn a random forest model and a representation simultaneously. The basic idea is to modify the representation so that subsequent trees in the random forest are less correlated.  The authors evaluate the technique empirically and show some modest gains. While the reviews were mixed, the approach is quite different from the usual approaches published at ICLR  and so I think it's worth highlighting this work.